SOFTWARE

# Revealing cancer driver genes through integrative transcriptomic and epigenomic analyses with Moonlight

Mona Nourbakhsh[1,2], Yuanning Zheng[3], Humaira Noor [3], Hongjin Chen[1], Subhayan Akhuli[1], Matteo Tiberti[2], Olivier Gevaert[3], Elena Papaleo[1,2]*

1 Cancer Systems Biology, Section for Bioinformatics, Department of Health Technology, Technical University of Denmark, Lyngby, Denmark, 2 Cancer Structural Biology, Danish Cancer Institute, Copenhagen, Denmark, 3 Department of Biomedical Data Science, Stanford Center for Biomedical Informatics Research, Palo Alto, California, United States of America

* elpap@dtu.dk, elenap@cancer.dk

## Abstract

Cancer involves dynamic changes caused by (epi)genetic alterations such as mutations or abnormal DNA methylation patterns which occur in cancer driver genes. These driver genes are divided into oncogenes and tumor suppressors depending on their function and mechanism of action. Discovering driver genes in different cancer (sub)types is important not only for increasing current understanding of carcinogenesis but also from prognostic and therapeutic perspectives. We have previously developed a framework called Moonlight which uses a systems biology multi-omics approach for prediction of driver genes. Here, we present an important development in Moonlight2 by incorporating a DNA methylation layer which provides epigenetic evidence for deregulated expression profiles of driver genes. To this end, we present a novel functionality called Gene Methylation Analysis (GMA) which investigates abnormal DNA methylation patterns to predict driver genes. This is achieved by integrating the tool EpiMix which is designed to detect such aberrant DNA methylation patterns in a cohort of patients and further couples these patterns with gene expression changes. To showcase GMA, we applied it to three cancer (sub)types (basal-like breast cancer, lung adenocarcinoma, and thyroid carcinoma) where we discovered 33, 190, and 263 epigenetically driven genes, respectively. A subset of these driver genes had prognostic effects with expression levels significantly affecting survival of the patients. Moreover, a subset of the driver genes demonstrated therapeutic potential as drug targets. This study provides a framework for exploring the driving forces behind cancer and provides novel insights into the landscape of three cancer sub(types) by integrating gene expression and methylation data.

**Data availability statement:** The Moonlight2R package, including the functionalities described in this paper, is available on Bioconductor (https://www.bioconductor.org/) and on GitHub (https://github.com/ELELAB/Moonlight2R). Data and scripts that have been used for the analyses described in this paper are available on GitHub (https://github.com/ELELAB/Moonlight2_GMA_case_studies) and on an Open Science Foundation (OSF) repository (https://osf.io/j4n8q/). The results published here are in whole or in part based upon data generated by The Cancer Genome Atlas (TCGA) Research Network: https://www.cancer.gov/tcga. Example data and vignette are available in S1 Data. The data used for this analysis are available at the Genomic Data Commons (https://portal.gdc.cancer.gov).

**Funding:** EP's group is supported by Elegant North (EN; Exploring Leukemia: Education Genetics And Technology; New Option for Rare diseases Towards Health), a collaboration between Oslo University Hospital, Capital Region of Denmark, Technical University Denmark (DTU), Abzu, Plesner, Region Skåne. EP's group is also supported by Danmarks Grundforskningsfond (Grant/Award Number: DNRF125), Hartmanns Fond (R241-A33877), LEO Foundation (LF17006), Novo-Nordisk Fonden Bioscience and Basic Biomedicine (NNF20OC0065262). Travel grants from Knud Højgaards Fond, William Demant Fonden, and Otto Mønsteds Fond were awarded to MN. OG's group was further supported by the National Cancer Institute (NCI) under award: R01 CA260271. The content is solely the responsibility of the authors and does not necessarily represent the official views of the National Institutes of Health. The funders had no role in study design, data collection and analysis, decision to publish, or preparation of the manuscript.

**Competing interests:** The authors have declared that no competing interests exist.

## Author summary

Cancer is a complex disease and a main cause of mortality worldwide. This heterogeneous disease arises due to accumulation of changes which occur in driver genes that drive cancer progression when they are altered. These driver genes are commonly divided into oncogenes, which promote cancer, and tumor suppressors, which prevent it. A major goal of cancer research is identifying these driver genes, crucial for increasing our current understanding of cancer biology and for developing novel treatment approaches. A large number of cancer driver genes have already been identified. However, the underlying mechanisms for the alterations in these genes is challenging to predict given their context-dependent behavior and the complexity of cancer. Such explanations are the focus of this study with the aim of providing evidence of why certain genes do not function normally in cancer. Within this context, we present a new functionality to our previously developed cancer driver predictive framework, Moonlight. This new functionality integrates multiple data types to predict oncogenes and tumor suppressors in a systems-biology-oriented manner that is freely available as a R package for the community.

## Introduction

Cancer is a complex and heterogeneous disease and a leading cause of death globally [1]. This widespread disease is categorized into multiple (sub)types and is characterized by stepwise accumulation of (epi)genetic alterations in cancer driver genes [2]. Driver genes are classified according to their function, i.e., oncogenes (OCGs) activated by gain-of-function mechanisms and tumor suppressor genes (TSGs) inactivated by loss-of-function mechanisms [3]. Recently, dual role genes also emerged which show context-dependent behavior and can act as both OCGs and TSGs in different biological contexts [4,5]. Driver genes participate in several cellular pathways conceptualized in the Hallmarks of Cancer, a collection of functional capabilities that cells gain during their transition from normal to tumor cells [6–8]. Distinct driver genes can initiate cancer development in different cancer types and even within subtypes of cancers originating from the same tissue. Thus, context-specific discovery of driver genes in light of the cancer hallmarks is essential. Numerous tools have been developed for prediction of driver genes based on varying computational methods which we recently reviewed [9]. Prediction of driver genes is essential for increasing current knowledge of cancer development and for analyzing and interpreting the vast amount of data in relation to cancer phenotypes which is important towards reversing these phenotypes, discovering novel drug targets, facilitating new treatment strategies, and designing precision medicine strategies [10–13]. We have contributed to this field with Moonlight which uses a multi-omics systems biology approach for prediction of driver genes [14,15].

The accumulated (epi)genetic alterations in driver genes include mutations, copy number variations, aberrant methylation levels, and histone modifications [3,16]. While abnormal methylation patterns are recognized as cancer-causing mechanisms, they have been described to a lesser extent compared to mutations [9]. Hypomethylation and hypermethylation, respectively representing loss and gain of methylation compared to normal conditions, have been described as activating and inactivating mechanisms of OCGs and TSGs, respectively [17–19]. For instance, Søes et al. found promoter hypomethylation and increased expression of putative OCG *ELMO3* to be associated with development of non-small cell lung cancer [20].

Here, we present a novel functionality of Moonlight2, expanding upon features presented in our previous work [15]. Specifically, we incorporate methylation evidence to Moonlight2 predicted driver genes as a source of epigenetic explanation of the deregulated expression of these genes. Information about methylation state is provided by EpiMix, an integrative tool for detecting aberrant DNA methylation patterns connected with expression changes in patient cohorts [21]. To showcase this new feature, we apply it to three cancer (sub)types (basal-like breast cancer, lung adenocarcinoma, and thyroid carcinoma) and discover driver genes in the context of cell proliferation and apoptosis, two well-established cancer hallmarks, and explore the prognostic and therapeutic potentials of the predicted driver genes. We apply our new method on data from The Cancer Genome Atlas (TCGA) [22,23].

## Design and implementation

### Design and implementation of a new functionality in Moonlight

Here, we present a new functionality to Moonlight, our framework for driver gene prediction [14,15]. Moonlight requires a set of differentially expressed genes (DEGs) as input and is built up of two layers (Fig 1A). In this context, a "layer" is made of a set of data analysis steps with a precise purpose. The primary layer uses gene expression differences between tumor and normal tissue, as well as information about cancer-related biological processes, to discover putative driver genes termed oncogenic mediators. This is done through four steps. The first step is a functional enrichment analysis (FEA) which assesses enrichment of cancer-related biological processes of the DEGs. This allows the user to understand the biological context in which the putative drivers will be predicted. Secondly, a gene regulatory network (GRN) analysis is carried out where interactions between the DEGs are modelled by means of mutual information. The resulting networks of DEGs are subsequently used to estimate the effect of the DEGs on the given biological processes through an upstream regulator analysis (URA). Finally, a pattern recognition analysis (PRA) is carried out where the DEGs are divided into putative OCGs and TSGs (termed oncogenic mediators) based on their effects on cancer growing and cancer blocking processes. For instance, if a DEG has a positive effect on cell proliferation and a negative effect on apoptosis, it would be categorized as a putative OCG and vice versa for putative TSGs [14,15]. Following this primary layer, a secondary layer couples mechanistic evidence to the oncogenic mediators by investigating (epi)genetic alterations in the oncogenic mediators (namely, mechanistic indicators). The rationale behind this is that gene expression changes alone are insufficient to explain the (in)activation of the drivers, hence a second layer of evidence incorporated in the (epi)genetic alterations is necessary. From this secondary layer, the critical driver genes are predicted among the oncogenic mediators (Fig 1A). We recently presented Moonlight2 with the overall goal of implementing new functionalities to provide standardized and automatized solutions to the analysis of the mechanistic indicators. At first, we developed a secondary layer for mechanistic indicators based on mutational data in a functionality called Driver Mutation Analysis (DMA) [15]. DMA first classifies mutations in the cancer cohort into driver and passenger mutations, and next retains those oncogenic mediators with at least one driver mutation as the final set of driver genes [15].

In this contribution, we added another secondary layer to Moonlight2 to cover mechanistic indicators related to methylation changes. This new functionality is termed Gene Methylation Analysis (GMA) and should be applied following the Pattern Recognition Analysis (PRA) function which predicts the oncogenic mediators in the primary layer (Fig 1A). To fully take advantage of the Moonlight framework, the user must apply a secondary layer following the primary layer, meaning

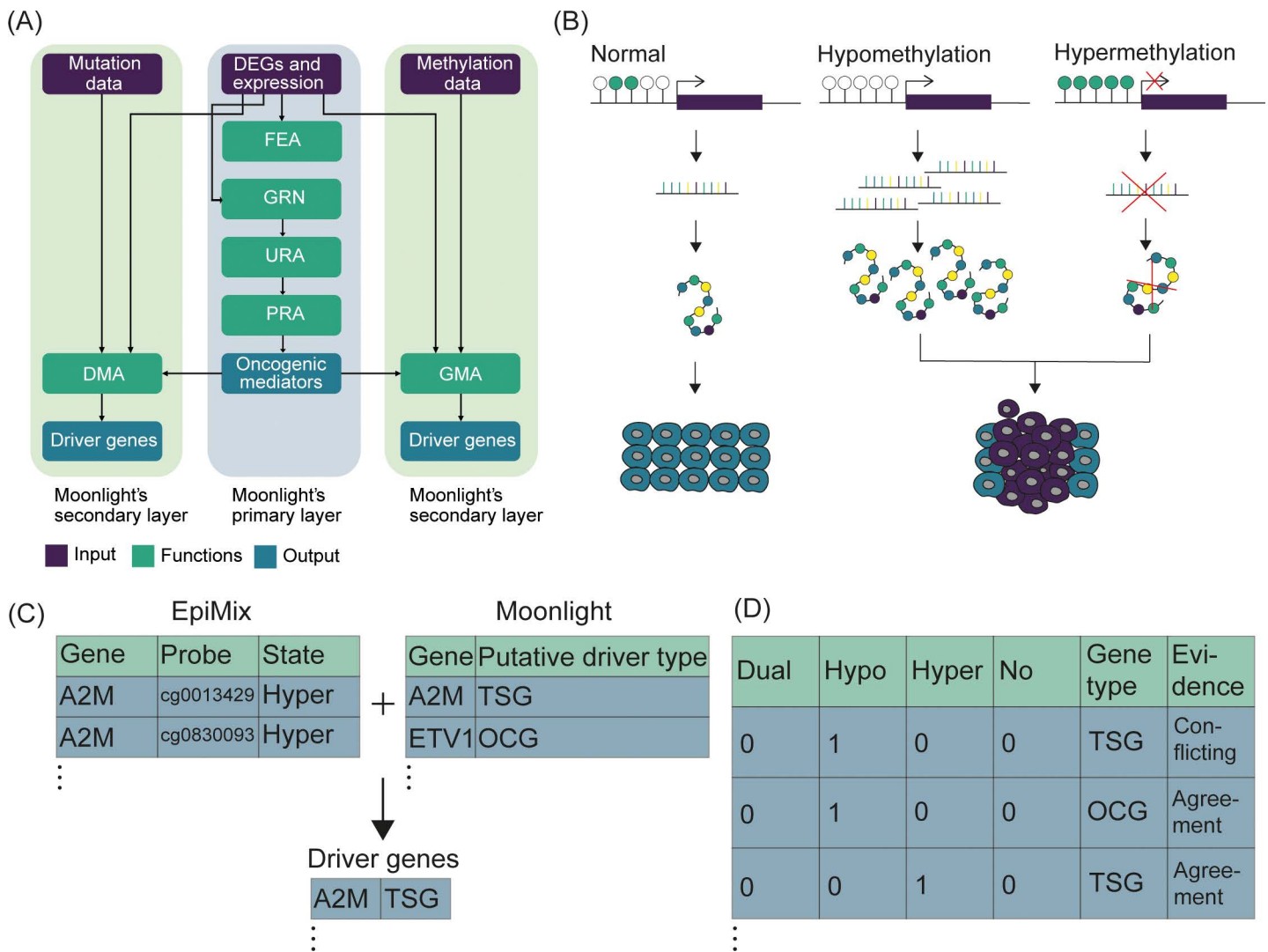

**Fig 1. Overview of the Moonlight framework with new methylation functionality.** (A) Moonlight consists of a primary layer requiring differentially expressed genes and gene expression data as input. The primary layer predicts oncogenic mediators through a series of functions called functional enrichment analysis (FEA), gene regulatory network analysis (GRN), upstream regulator analysis (URA), and pattern recognition analysis (PRA). Moonlight's secondary mutation layer requires mutation data as input and is carried out via the driver mutation analysis (DMA) function and similarly, Moonlight's secondary methylation layer implemented in the gene methylation analysis (GMA) function requires methylation data as input. The secondary layer results in the final prediction of driver genes. (B) DNA methylation is a mechanism occurring under physiological conditions in cells which functions to regulate gene expression. However, in cancer, the DNA methylation process is altered. A loss of methylation called hypomethylation can occur which can lead to increased expression of a gene and thus an increased amount of the resulting protein. In contrast, gain of methylation called hypermethylation can also occur which can silence gene expression and lead to decreased protein expression. These two mechanisms can finally lead to cancer. Hypo- and hypermethylation can activate and inactivate oncogenes and tumor suppressors, respectively, the biological principle that GMA is built on. (C) The outputs of EpiMix and Moonlight are integrated to predict driver genes. EpiMix outputs a table of CpG-gene pairs containing differentially methylated CpG sites whose DNA methylation state is associated with gene expression. Moonlight outputs a list of oncogenic mediators and their putative driver role as tumor suppressors or oncogenes. (D) Driver genes are defined in GMA by comparing EpiMix's predictions of methylation state and Moonlight's predictions of driver role in "evidence" categories. Those oncogenic mediators labeled with an "agreement" evidence are retained as the final set of predicted driver genes.

PLOS Computational Biology

the user must apply either DMA or GMA depending on the research question and available source of -omics data. The user can also apply both DMA and GMA if this is of interest, with the expectation that they would provide complementary evidence since they take into consideration different sources that can affect gene expression. The biological foundation for GMA lies within the observed roles of DNA methylation in both physiological and cancer states. Under healthy conditions, DNA methylation serves an essential regulatory role in cells by regulating expression of genes [24]. However, in cancer, DNA methylation processes are altered, where hypo- and hypermethylation can activate and inactivate OCGs and TSGs, respectively, leading to overexpression of OCGs and silencing of TSGs [17–19] (Fig 1B).

GMA predicts methylation-driven driver genes by using EpiMix [21]. EpiMix models DNA methylation in patient cohorts and predicts differential methylation associated with gene expression and further allows for DNA methylation analysis of non-coding regulatory regions [21], therefore being perfectly suitable to complement Moonlight's primary layer. Here, we are using EpiMix's "regular mode" as this allows for analyzing DNA methylation in the promoter regions. In brief, EpiMix first uses a beta mixture model to decompose the DNA methylation profiles of the cohort. Next, differential methylation values are calculated, which represent the mean differences in DNA methylation levels between patients in each of the identified mixture components in the disease group compared with the control group. Finally, EpiMix finds those CpG sites that are significantly associated with gene expression [21]. EpiMix is available as a R BioConductor package, which allows for easy integration with Moonlight. A key result of EpiMix is a table which includes functional CpG-gene pairs containing differentially methylated CpG sites whose DNA methylation state is associated with the expression of the corresponding genes they map to. Moreover, the methylation state (e.g., hypo- or hypermethylated) of each CpG site is reported. This table is integrated with the main output table from Moonlight's primary layer, specifically the output from PRA, which provides a list of oncogenic mediators and their putative driver role (e.g., putative TSG or OCG) (Fig 1C). This integration step involves the following: for each oncogenic mediator, the number of associated CpG sites is summarized. EpiMix's predictions of methylation state and Moonlight's predictions of driver gene role are then compared and used to assess whether the gene's methylation status supports the putative role (OCG or TSG) of the oncogenic mediator. These comparisons are subsequently used to define the driver genes based on the assumption with which methylation changes can activate or inactive driver genes (Fig 1D, Table 1). For example, suppose EpiMix predicts a gene to contain one or more hypermethylated CpG sites whereas Moonlight's primary layer predicts this gene to be an OCG. This indicates conflicting evidence between the mechanism of (in)activation and putative driver role between the two tools. Consequently, this gene is labeled with "conflicting" evidence. Contrary, if EpiMix for instance predicts a gene to be associated with hypermethylated CpG site(s) and Moonlight predicts this gene as a TSG, this gene is labeled with an "agreement" evidence to signify correspondence between EpiMix and Moonlight (Fig 1D, Table 1). Those oncogenic mediators labeled with an "agreement" evidence are retained as the final set of predicted driver genes. Those oncogenic mediators labeled with a "conflicting" evidence are still included in one of the outputs of GMA, but they are not retained as part of the predicted driver genes set.

**Table 1. Comparison between EpiMix's predictions of methylation state and Moonlight's primary layer's predictions of driver gene role in "evidence" categories. Those oncogenic mediators labeled with "agreement" evidence are retained as the final set of predicted driver genes.**

| Methylation status (EpiMix) | Putative driver type (Moonlight's primary layer) | Evidence category |
|---|---|---|
| Hypomethylation | OCG | Agreement (OCG) |
| Hypomethylation | TSG | Conflicting |
| Hypermethylation | OCG | Conflicting |
| Hypermethylation | TSG | Agreement (TSG) |

Abbreviations: OCGs, oncogenes; TSGs, tumor suppressor genes.

As input, GMA requires i) a gene expression matrix with genes in rows and tumor and normal samples in columns, ii) a methylation matrix with CpG sites in rows and tumor and normal samples in columns which should be the same samples as in the expression data, iii) output of PRA from Moonlight's primary layer, i.e., the predicted oncogenic mediators and their putative driver role, iv) output of a differential expression analysis (DEA) which includes information about the DEGs, and finally, v) a table containing information about the samples which includes the sample names and the sample types (tumor or normal sample). In return, GMA outputs the following: i) a list of predicted driver genes categorized into TSGs and OCGs, ii) a summary of the oncogenic mediators which includes the number of associated CpG sites and evidence label, iii) a summary of various annotations found to all DEGs input to Moonlight on the gene and methylation level, and iv) raw EpiMix results corresponding to applying EpiMix on the input data independently of the GMA function.

We have also created three functions for visualizing genes and methylation states: plotGMA which visualizes the number of differentially methylated hypo-, hyper- or dual-methylated CpG sites, plotMoonlightMet which visualizes the effect of genes on biological processes estimated in Moonlight's primary layer, and plotMetExp which calls a visualization function from EpiMix, EpiMix_PlotModel, to display gene expression and methylation levels of a specific gene and CpG site [21].

### Application of new functionality to three cancer (sub)types

Following implementation of the new functionality, GMA, in Moonlight2, we conducted a case study applying GMA to basal-like breast cancer, lung adenocarcinoma, and thyroid carcinoma data from TCGA to discover methylation-driven driver genes. Moreover, we compared these predicted drivers with mutation-driven drivers by applying our previously developed secondary mutational layer called DMA [15]. We selected these cancer types to compare with and build upon our previous findings where we examined mutational drivers with the DMA functionality [15]. Detailed methods behind this case study are included in S1 Text.

## Results

### Case study: Prediction of driver genes with differential methylation in three different cancer types using Moonlight2

To showcase the new functionality in Moonlight2 and predict driver genes driven by methylation changes, we applied Moonlight2 on three cancer (sub)types: basal-like breast cancer, lung adenocarcinoma, and thyroid carcinoma. First, we used RNAseq data to perform DEA between each of these cancer tissues and corresponding normal samples as this is the input to Moonlight's primary layer (Table 2). Following DEA, Moonlight's primary layer predicted 159, 1228, and 1598 oncogenic mediators in these three cancer (sub)types, respectively (Table 2). Additionally, EpiMix alone identified 9483, 10018, and 6142 functional gene-CpG pairs in these three cancer (sub)types, respectively. These functional gene-CpG pairs represent differentially methylated CpG sites whose DNA methylation state is associated with the expression of the corresponding genes they map to. The number of hits discovered individually from EpiMix and Moonlight's primary layer indicate a substantial amount of significant associations. Consequently, integrating the results from EpiMix with the

Table 2. Number of predicted DEGs, oncogenic mediators, and driver genes in three cancer (sub)types: basal-like breast cancer, lung adeno-carcinoma, and thyroid carcinoma. The oncogenic mediators and driver genes predicted by Moonlight's primary and secondary methylation layer, respectively, are divided into (putative) TSGs and OCGs.

| Cancer (sub)type | DEGs | Oncogenic mediators [putative TSGs/putative OCGs] | Driver genes [TSGs/OCGs] |
|---|---|---|---|
| Basal-like breast cancer | 4292 | 159 [125/34] | 33 [32/1] |
| Lung adenocarcinoma | 4468 | 1228 [521/707] | 190 [110/80] |
| Thyroid carcinoma | 2972 | 1598 [118/1480] | 263 [5/258] |

Abbreviations: DEGs, differentially expressed genes; OCGs, oncogenes; TSGs, tumor suppressor genes.

oncogenic mediators identified in Moonlight's primary layer, as implemented in GMA presented here, helps narrowing down the dataset to the most relevant findings in a synergistic fashion. From GMA, we found that those oncogenic mediators in basal-like breast cancer that are associated with differentially methylated CpGs include in total 38 hypomethylated CpGs, 165 hypermethylated CpGs, and 22 methylated CpGs with a dual status, the latter meaning the CpG site was found hypomethylated in cancer tissues from some patients, while hypermethylated in other patients. Similarly, oncogenic mediators in lung adenocarcinoma that are associated with differentially methylated CpGs include in total 218 hypomethylated CpGs, 625 hypermethylated CpGs, and 48 dual-methylated CpGs. Finally, oncogenic mediators in thyroid carcinoma associated with differentially methylated CpGs contain in total 945 hypomethylated CpGs, 305 hypermethylated CpGs, and 230 dual-methylated CpGs (Fig 2A).

Across all three cancer (sub)types, we found that the largest number of differentially methylated CpG sites mapping to a single oncogenic mediator was 28. The classifications of methylation status in the oncogenic mediators in basal-like breast cancer are shown in Fig 2B, generated with the plotGMA function. Next, we compared Moonlight's oncogenic mediators with EpiMix' functional genes. For this, we included only those functional genes that contained the same methylation state in all its associated CpGs and moreover, the genes with a dual methylation state, as previously defined, were excluded. In basal-like breast cancer, this comparison revealed 109 oncogenic mediators not associated with differentially methylated CpGs, 17 oncogenic mediators with a "conflicting" evidence label, and 33 oncogenic mediators with an "agreement" evidence label (Fig 2C). Consequently, these 33 oncogenic mediators are retained as the final set of driver genes divided into 32 TSGs and 1 OCG (Table 2). Next, we visualized the effect of these predicted driver genes in basal-like breast cancer on two well-known cancer hallmarks, apoptosis and proliferation of cells, using the function plotMoonlight-Met. These effects define the basis upon which the oncogenic mediators are predicted from the PRA step in Moonlight's primary layer, demonstrating that the predicted OCGs have a positive effect on proliferation of cells and a negative effect on apoptosis and vice versa for the predicted TSGs (Fig 2D). Similar overviews for lung adenocarcinoma and thyroid carcinoma are shown in S1 Fig, which resulted in a final prediction of 190 driver genes divided into 110 TSGs and 80 OCGs in lung adenocarcinoma and 263 driver genes categorized into 5 TSGs and 258 OCGs in thyroid carcinoma (Table 2). We did not discover any dual role genes across the three cancer (sub)types, i.e., genes predicted as OCGs in one of the three cancer (sub)types and as TSGs in another cancer (sub)type and vice versa.

We then compared the predicted driver genes with the predicted oncogenic mediators in each cancer (sub)type. We quantified these comparisons in terms of overlaps with genes reported in the COSMIC Cancer Gene Census (CGC) [25]. Specifically, we computed the precision as (TP/(TP+FP))*100 and sensitivity as (TP/(TP+FN))*100. We defined the true positives (TP) as the overlap between the gene set (either the driver genes or the oncogenic mediators) and the CGC, whereas the false positives (FP) are those genes found in the gene set but are not included in CGC. In contrast, the false negatives (FN) comprise those genes reported in CGC but are not predicted in our gene set. For all three cancer (sub) types, we found that GMA had a greater precision and lower sensitivity compared to using only Moonlight's primary layer (Fig 2E). A higher precision of GMA is desirable as it indicates that the predicted driver gene sets have a higher fraction of genes from the CGC compared to the oncogenic mediator sets. On the other hand, the higher sensitivity of using only Moonlight's primary layer compared to also using GMA might be attributed to the larger numbers of oncogenic mediators. A larger number of oncogenic mediators results in a larger overlap between the CGC and the oncogenic mediators, thereby lowering the number of FNs and increasing the sensitivity. In this case, prioritizing higher precision over sensitivity is preferable since our aim is to find the most crucial driver genes among the oncogenic mediators. Thus, a higher precision indicates a greater proportion of TPs, corresponding with our objective. Next, we also evaluated the significance of association between the gene sets and the CGC using a Fisher's exact test (Table 3). We only found the oncogenic mediator and driver gene sets from basal-like breast cancer to have a significant association with genes in the CGC ($p$-value=0.000392 for the oncogenic mediators predicted using Moonlight's primary layer and $p$-value=0.00228 for the driver genes predicted using GMA). However, in all three cancer (sub)types, we found the driver genes to have a higher

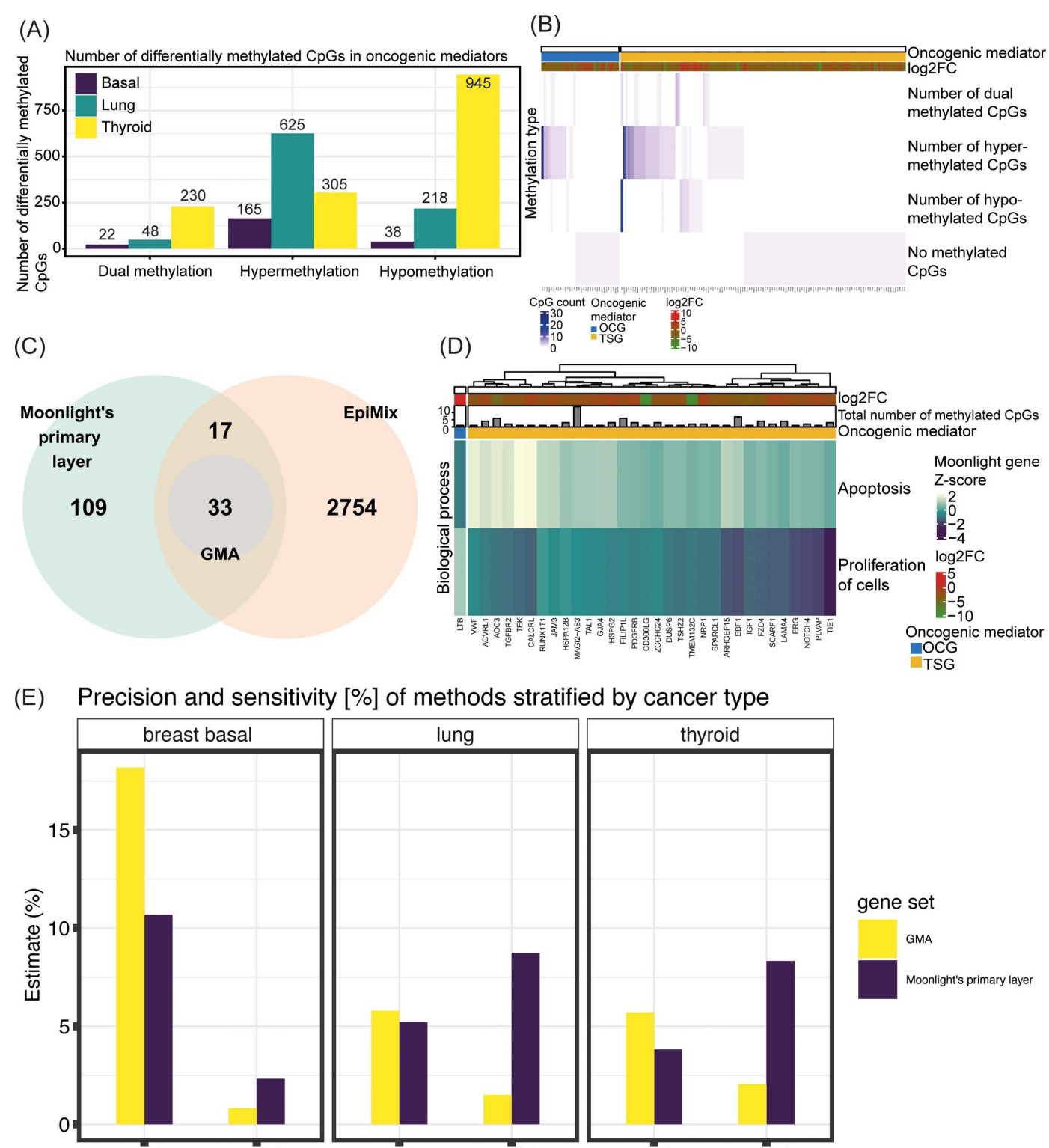

**Fig 2. Integration of Moonlight and EpiMix for prediction of cancer driver genes.** (A) Number of differentially methylated CpGs as found from EpiMix in oncogenic mediators predicted from Moonlight's primary layer. The differentially methylated CpGs are categorized into methylation status and

stratified by cancer (sub)type. (B) Heatmap showing number of differentially methylated CpGs and classifications of methylation status in the oncogenic mediators in basal-like breast cancer. The heatmap was generated using the plotGMA function. (C) Venn diagram comparing oncogenic mediators predicted from Moonlight's primary layer with functional genes predicted from EpiMix in basal-like breast cancer. The functional genes are genes containing differentially methylated CpG pairs whose DNA methylation state is associated with expression of the gene. Only those functional genes that contained the same methylation state in all of its associated CpGs were included in this comparison, and moreover, the dual methylation states were excluded. (D) Heatmap showing the effect of the predicted driver genes in basal-like breast cancer on apoptosis and proliferation of cells. This heatmap was generated using the function plotMoonlightMet. These effects define the basis upon which the oncogenic mediators are predicted from the PRA step in Moonlight's primary layer. (E) Comparison between the predicted driver genes with the predicted oncogenic mediators in all three cancer (sub)types where the driver genes were predicted with the new functionality GMA in Moonlight's secondary layer, and the oncogenic mediators were predicted with Moonlight's primary layer. The comparisons were quantified in terms of overlaps with genes reported in the COSMIC Cancer Gene Census (CGC) by computing the precision and sensitivity. The precision was calculated as (TP/(TP+FP))*100 and sensitivity as (TP/(TP+FN))*100. The true positives (TP) are the overlap between the gene set (either the driver genes or the oncogenic mediators) and the CGC. The false positives (FP) are those genes found in the gene set but are not included in CGC. The false negatives (FN) comprise those genes reported in CGC but are not predicted in our gene set.

**Table 3. Significance of association between Moonlight's gene sets and genes from the Cancer Gene Census (CGC) evaluated using Fisher's exact test in three cancer (sub)types: basal-like breast cancer, lung adenocarcinoma, and thyroid carcinoma. The gene sets from Moonlight were found using Moonlight's primary layer and Moonlight's secondary layer through the Gene Methylation Analysis (GMA) functionality. p-values and odds ratios from Fisher's exact test are includdned.**

| Cancer (sub)type | Method | *p*-value | Odds ratio |
|---|---|---|---|
| **Basal-like breast cancer** | Moonlight's primary layer | 0.000392* | 2.77 |
| | GMA | 0.00228* | 5.09 |
| **Lung adenocarcinoma** | Moonlight's primary layer | 0.0764 | 1.28 |
| | GMA | 0.272 | 1.40 |
| **Thyroid carcinoma** | Moonlight's primary layer | 0.472 | 0.895 |
| | GMA | 0.215 | 1.39 |

*$p$-value$<0.05$

Abbreviations: CGC, Cancer Gene Census; GMA, Gene Methylation Analysis.

odds ratio than the oncogenic mediators, demonstrating a greater association between the driver gene sets and the CGC compared to the oncogenic mediators (Table 3).

While these results together demonstrate the added value of GMA, it is worth highlighting certain limitations. Notably, the driver genes reported in CGC are mainly based on mutation evidence. In this study, we have used abnormal DNA methylation levels as evidence for deregulated expression of the driver genes. Hence, these methylation patterns may not be fully captured in the CGC, challenging our comparison with the CGC. However, to date, no golden standard of cancer drivers exists, and the CGC stands as the most robust and comprehensive resource available. Thus, it serves as the main reference point that most studies use to evaluate their predicted driver genes and method [26–37]. To our knowledge, a similar well-curated resource of cancer driver genes driven by methylation changes does not exist. Moreover, performing cancer type-specific comparisons would be more desirable. While the CGC reports which cancer types the driver genes are associated with, these annotations are limited in scope. Therefore, while ideal, performing such cancer type-specific comparisons do not contain enough power. Finally, the quantitative statistical measures are not taking into account that some of our predicted driver genes may be novel. Consequently, some FPs may in fact be TPs but are not included in CGC, and some FNs may not necessarily be FNs; rather, they may not represent drivers in the specific cancer (sub)type.

To investigate biological roles of the predicted driver genes, we performed enrichment analyses (Fig 3). The predicted driver genes are involved in various signaling pathways such as KRAS signaling in basal-like breast cancer and thyroid carcinoma, mTORC1 signaling in lung adenocarcinoma, and TNF−alpha signaling via NF−kB and p53 pathway in thyroid carcinoma. Previously, TP53 and TNF signaling have been associated with the onset of cancer among epigenetically modified

 

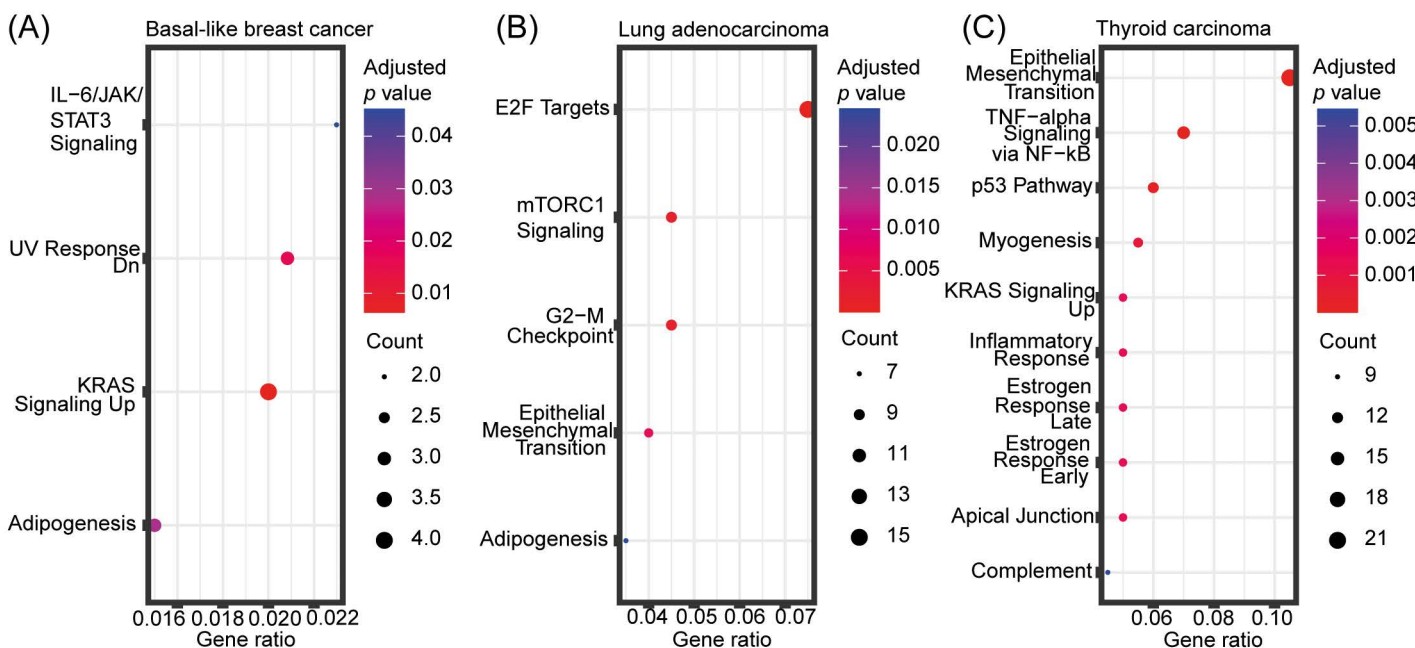

**Fig 3. Enrichment analyses of predicted driver genes.** Enrichment analysis of predicted driver genes in **(A)** basal-like breast cancer, **(B)** lung adenocarcinoma, and **(C)** thyroid carcinoma using the "MSigDB Hallmark 2020" database. The top 10 most significantly enriched terms (adjusted *p*-value < 0.05) are included. The gene ratio on the *x* axis is the ratio between the number of predicted driver genes that intersect with genes annotated in the given hallmark gene set and the total number of genes annotated in the respective hallmark gene set. The point sizes reflect the number of driver genes playing a role in the respective hallmark gene set.

pathways [38]. Furthermore, IL−6/JAK/STAT3 signaling was significantly enriched among the predicted driver genes in basal-like breast cancer (Fig 3). Basal-like breast cancers overexpress Interleukin 6 (IL-6), a pro-inflammatory cytokine, and it has been reported that p53 absence triggers an IL-6 dependent epigenetic reprogramming driving breast cancer cells towards a basal-like/stem cell-like gene expression profile [39]. Additionally, epithelial-mesenchymal transition (EMT) is a recurring enriched term, observed in both lung adenocarcinoma and thyroid carcinoma. Epigenetic regulation of EMT has previously been described, and DNA methylation and demethylation plays a key role in this regulation [40–43].

## Association between expression of predicted driver genes and survival of cancer patients

We performed survival analysis to evaluate the prognostic potential of the predicted driver genes. We first used Cox proportional hazards regression and found that the expression level of 20 of the predicted OCGs in lung adenocarcinoma had a significant effect on survival at the multivariate level when accounting for tumor stage, age of patients, and sex of patients. Similarly, expression of two of the predicted OCGs in thyroid carcinoma had a significant effect on survival. No driver genes in basal-like breast cancer were found to have a significant effect on survival at the multivariate level. Thus, we deemed these 22 OCGs as prognostic (Fig 4A). Next, we examined whether high or low expression of these prognostic genes were associated with survival of the patients. For this, we divided the patients into high and low expression groups and assessed differences in survival through Kaplan-Meier survival analyses and log-rank tests. These analyses revealed a significant difference in survival between patients with high and low expression of 18 of the 20 prognostic OCGs in lung adenocarcinoma. The two OCGs that did not show a significant difference were *RPL39L* and *GINS2*. On the other hand, we did not observe a significant difference in survival between patients with high and low expression of the two predicted OCGs in thyroid carcinoma. These results together indicate a greater prognostic potential of OCGs

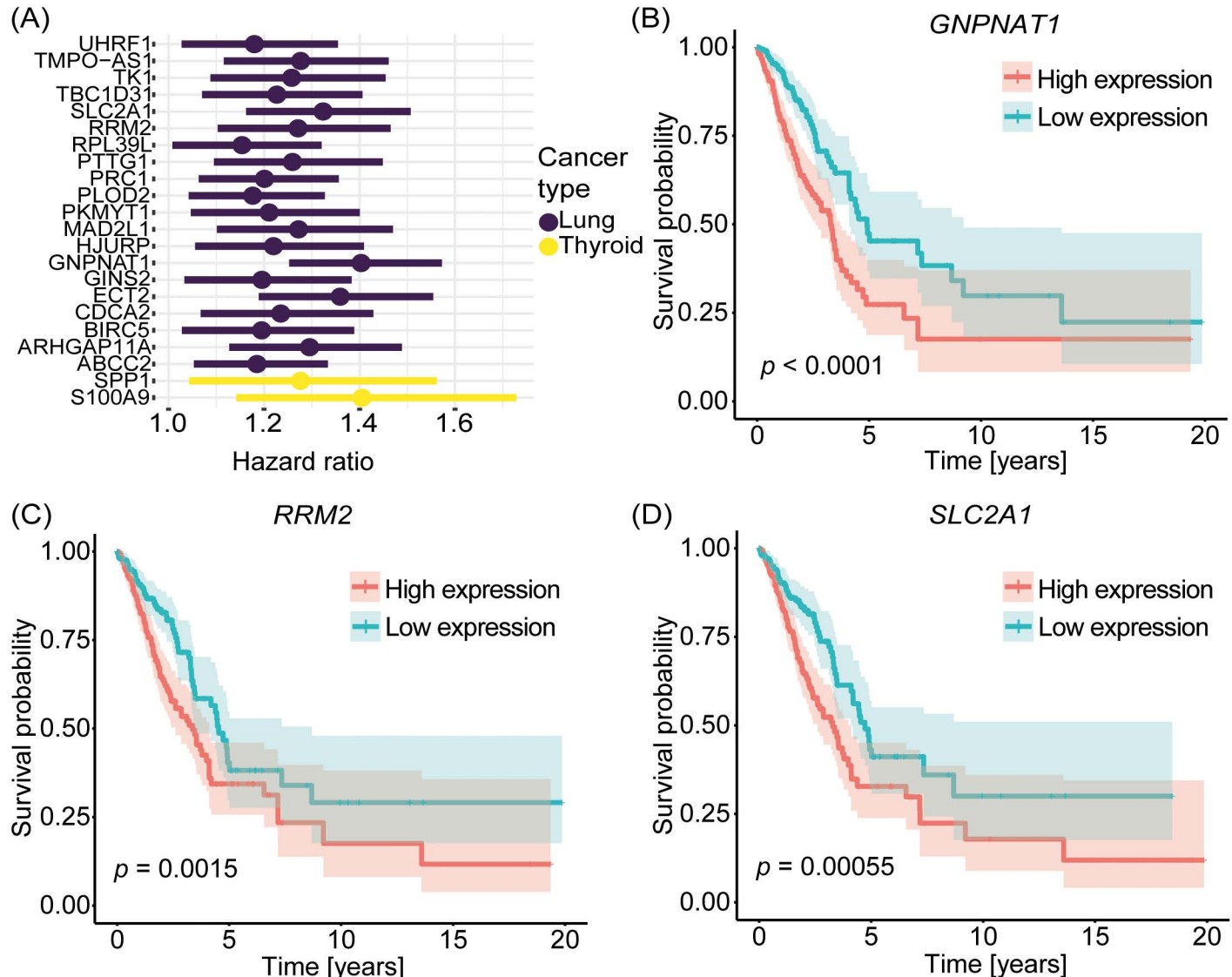

**Fig 4. Survival analysis of predicted driver genes.** (A) Hazard ratios from multivariate Cox proportional hazards regression of 20 of the predicted OCGs in lung adenocarcinoma and of two of the predicted OCGs in thyroid carcinoma. (B-D) Kaplan-Meier survival plots of three of the predicted OCGs in lung adenocarcinoma which were deemed prognostic from multivariate Cox regression analysis: (B) *GNPNAT1*, (C) *RRM2*, and (D) *SLC2A1*. Patients with expression values above and below the median expression level of the respective gene were divided into a high and low expression group, respectively. The *p*-values represent the significance of difference in survival between the two groups for each gene.

compared to TSGs and additionally, a greater presence of prognostic OCGs in lung adenocarcinoma compared to basal-like breast cancer and thyroid carcinoma. It is, however, worth mentioning that a smaller subset of driver genes was predicted in basal-like breast cancer with only one predicted OCG, indicating a more limited search pool for prognostic OCGs. The Cox regression analysis demonstrated that increases in expression of 20 OCGs in lung adenocarcinoma and two OCGs in thyroid carcinoma are associated with an increase in the hazard of experiencing death. However, while the Kaplan-Meier analyses showed statistically significant differences between high and low expression groups of 18 of the 20 prognostic OCGs in lung adenocarcinoma, the magnitudes of the survival differences are small. Small absolute survival

differences can challenge clinical implications, and further studies are needed in larger cohorts to determine prognostic potential of the driver genes.

To highlight a few examples, multivariate Cox regression analysis identified *GNPNAT1*, *RRM2*, and *SLC2A1* as prognostic OCGs in lung adenocarcinoma with hazard ratios of 1.4, 1.3, and 1.3, respectively. In all three cases, patients with high expression of the OCG had a significantly lower survival probability compared to patients with low expression of these OCGs (Fig 4B-D) (*p*-values: < 0.0001, 0.0015 and 0.00055 for *GNPNAT1*, *RRM2* and *SLC2A1*, respectively). This aligns with the anticipated role of OCGs which are typically upregulated in cancer, indicating a worse prognosis.

## Predicted driver genes have therapeutic potential as drug targets

The potential of cancer driver genes as drug targets have previously been highlighted [44–46] and targeted therapies have been developed towards these genes. Thus, we next investigated the therapeutic potential of the predicted driver genes as drug targets by querying the Drug-Gene Interaction Database (DGIdb) [47] for driver gene-drug interactions using only cancer-specific data sources (see also S2 Text). In basal-like breast cancer, we identified seven TSGs documented to interact with drugs in DGIdb. In lung adenocarcinoma, both OCGs and TSGs, numbering 12 each, were reported as drug targets. Finally, in thyroid carcinoma, 23 OCGs were reported as interacting with drugs (S2 Fig). Across all three cancer (sub)types, the number of driver gene-drug interactions varied between one and 55. Roughly half of all predicted driver genes interacted with one drug while the other half interacted with two or more drugs (Fig 5A).

Next, we examined those driver gene-drug interactions for which the interaction type was known. In basal-like breast cancer, lung adenocarcinoma, and thyroid carcinoma, we found three (all TSGs), six (three OCGs and three TSGs), and five (all OCGs) driver genes, respectively, for which the interaction type was known (Fig 5B-D). Most of the drugs were classified as inhibitors. The two driver genes with the most interactions were *PDGFRB* in basal-like breast cancer and *MET* in thyroid cancer. We predicted *PDGFRB* as a TSG in basal-like breast cancer which is annotated to interact with 16 inhibitors and three drugs with antagonist or inhibitor interactions. These drugs exert inhibitory mechanisms for targeting an OCG role of *PDGFRB*. As the gene-drug target interactions are not specific for a certain cancer type, these results might suggest a potential dual role of *PDGFRB*. On the other hand, *MET* predicted as an OCG in thyroid cancer interacted with 19 inhibitors, in accordance with the OCG role of *MET*. Moreover, in lung adenocarcinoma, the predicted OCG *RRM2*, which we also identified as a prognostic gene above, interacted with one inhibitor, gemcitabine. Previously, one study investigated the mRNA expression of RRM1 and RRM2 in tumors from patients with lung adenocarcinoma treated with docetaxel/gemcitabine. They found low RRM2 mRNA expression to be associated with a higher response rate to treatment compared to patients with high expression [48]. Similarly, in thyroid carcinoma, we observed an interaction between *ERBB3*, a member of the epidermal growth factor receptor (EGFR) family of receptor tyrosine kinases, and four inhibitors (sapitinib, poziotinib, gefitinib, and dacomitinib). These inhibitors, all classified as tyrosine kinase inhibitors [49–56], align with *ERBB3*'s predicted role as an OCG. Another example is the interaction between *EpCAM* and solitomab in lung adenocarcinoma. *EpCAM* is an epithelial cell adhesion molecule which plays a role in cell proliferation, migration, and signaling and is frequently overexpressed on the cell surface of several human carcinomas [57–59]. For instance, *EpCAM* was recently found to be upregulated in primary lung cancer compared to normal lung tissues caused by gene amplification and promoter hypomethylation [60]. Solitomab is a bispecific antibody binding to *EpCAM* and *CD3* [57] which previously has shown preliminary signs of antitumor activity [61].

## Integrating the results from DMA and GMA functions of Moonlight2

Next, we also applied the Moonlight2 DMA functionality [15] to the data used for the case studies above to show the potential of integrating different mechanistic indicators. For basal-like breast cancer, DMA predicted 46 driver genes (10 OCGs and 36 TSGs), while GMA predicted 33 driver genes (32 OCGs and 1 TSG) (Fig 6A-C). For lung adenocarcinoma, DMA predicted 842 driver genes (490 OCGs and 352 TSGs), while GMA predicted 190 (80 OCGs and 110 TSGs) (Fig

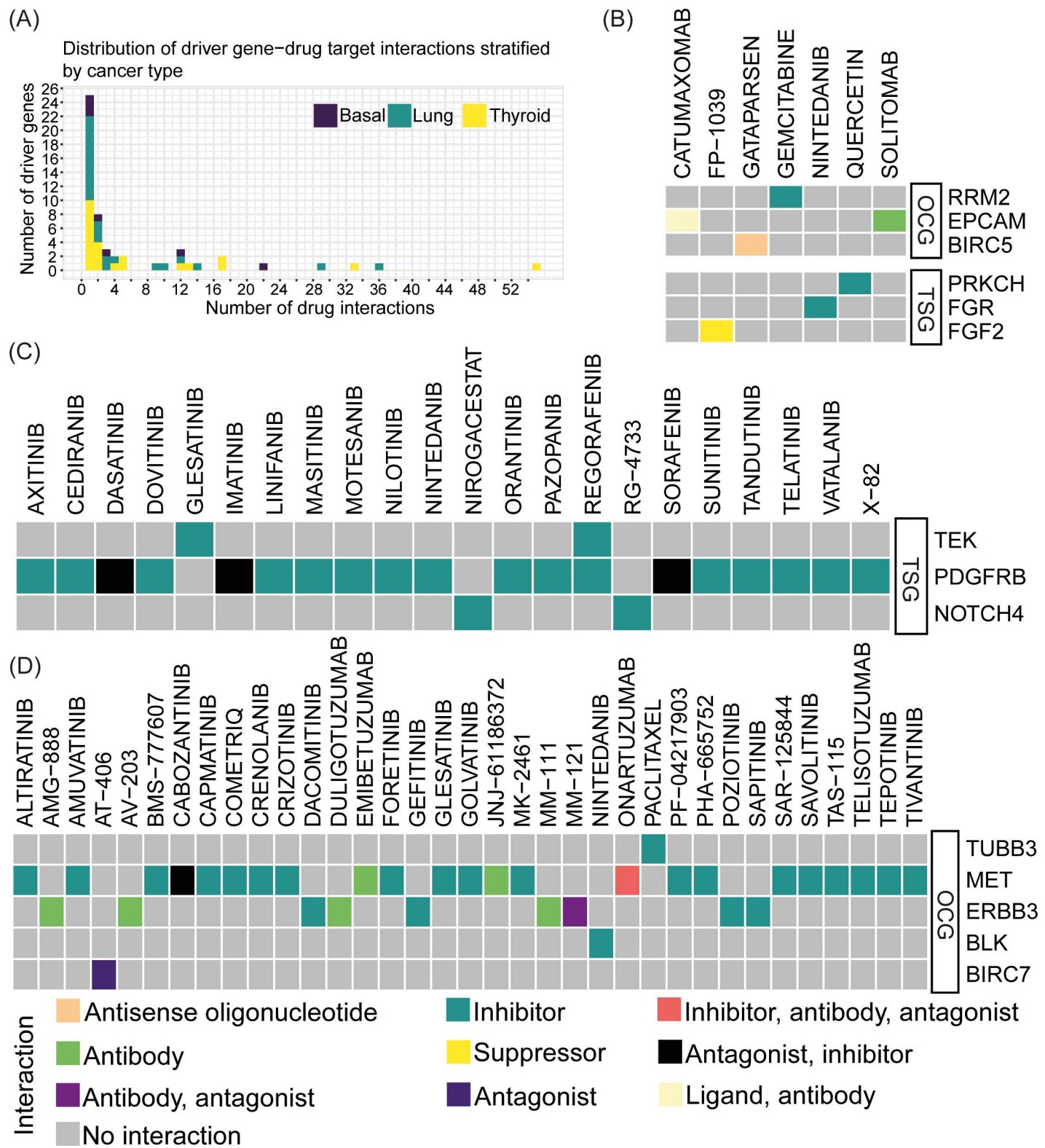

**Fig 5. Exploration of predicted driver genes as drug targets.** (A) Distribution of driver gene-drug interactions stratified by cancer type with the number of drug interactions on the *x* axis and number of driver genes on the *y* axis. (B-D) Heatmaps visualizing driver gene-drug interactions in **(B)** lung adenocarcinoma, **(C)** basal-like breast cancer, and **(D)** thyroid carcinoma. Only those driver gene-drug interactions where the interaction type was known are included in the heatmaps. The type of interaction is shown in different colors. The driver genes are divided into OCGs and TSGs.

PLOS Computational Biology | https://doi.org/10.1371/journal.pcbi.1012999    April 21, 2025

## Basal-like breast cancer

(A)
### Predicted driver genes

(B)
### Predicted tumor suppressor genes

(C)
### Predicted oncogenes

## Lung adenocarcinoma

(D)
### Predicted driver genes

(E)
### Predicted tumor suppressor genes

(F)
### Predicted oncogenes

**Fig 6. Comparison of number of mutation- and methylation-driven driver genes.** Venn diagram comparing **(A, D)** the number of driver genes, (B, E) TSGs, and (C, F) OCGs predicted by the Driver Mutation Analysis (DMA) and Gene Methylation Analysis (GMA) functions of Moonlight2 for **(A-C)** basal-like breast cancer and **(D-F)** lung adenocarcinoma.

6D-F). Both secondary layers predicted a larger number of driver genes in lung adenocarcinoma than basal-like breast cancer (Table 2, Fig 6). This is likely a direct consequence of Moonight's primary layer, which identified a larger number of oncogenic mediators in lung adenocarcinoma than basal-like breast cancer. At the same time, DMA predicted a larger number of driver genes for both datasets than GMA, with a larger proportion in lung adenocarcinoma than basal-like breast cancer (~4.5 times against ~1.4 times, respectively). This observation aligns with previous reports suggesting that lung adenocarcinoma exhibits a high mutation burden [62,63], suggesting that DMA was able to identify a larger number of driver mutations overall. In most cases, we found an overlap between driver genes identified by DMA and GMA, which suggests multiple mechanisms at play. In basal-like breast cancer, 13 driver genes were predicted by both DMA and GMA,

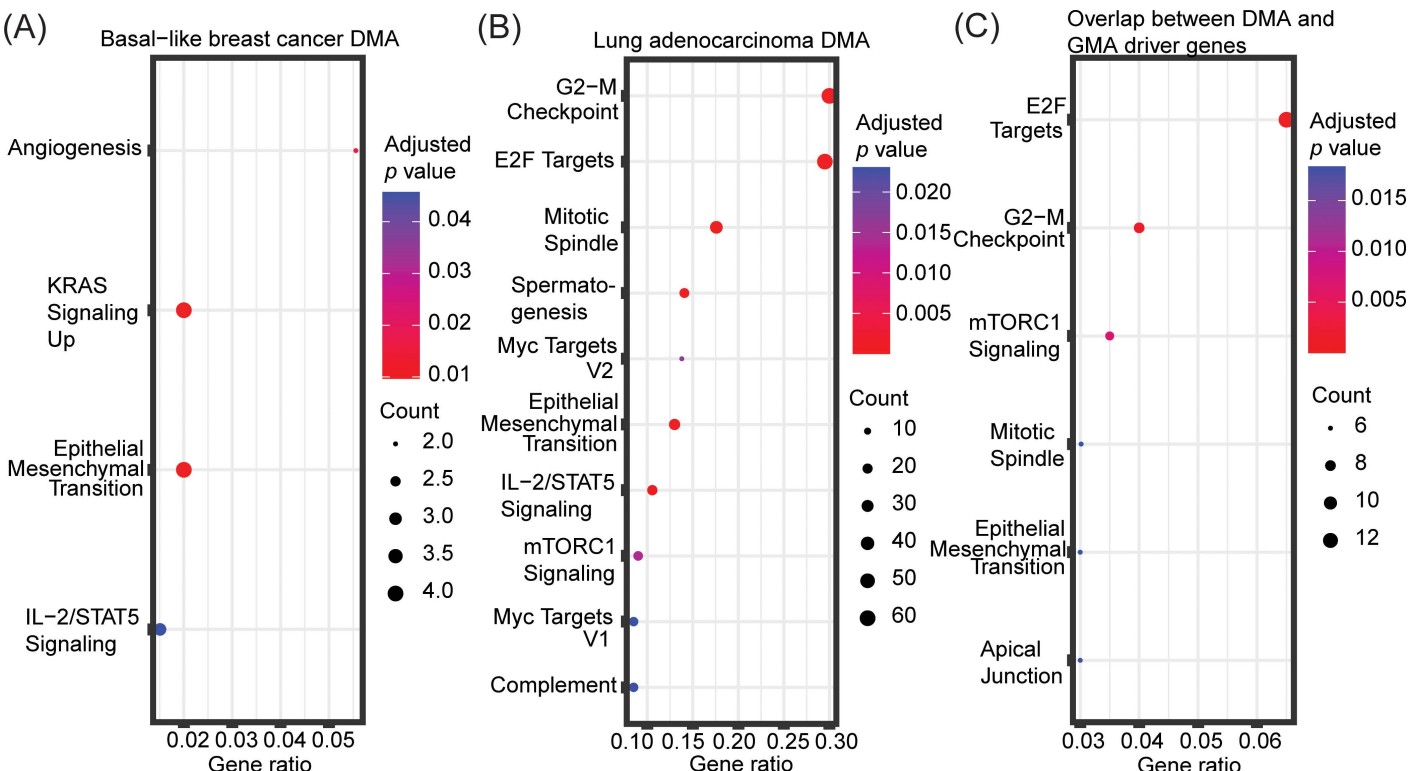

**Fig 7. Enrichment analysis of predicted mutation-driven driver genes.** Enrichment analysis of **(A)** mutation-driven driver genes predicted by Driver Mutation Analysis (DMA) in basal-like breast cancer, **(B)** mutation-driven driver genes predicted by Driver Mutation Analysis (DMA) in lung adenocarcinoma, and **(C)** driver genes predicted by both DMA and GMA in lung adenocarcinoma. The "MSigDB Hallmark 2020" database was used for the enrichment analyses. The top 10 most significantly enriched terms (adjusted $p$-value < 0.05) are included. The gene ratio on the $x$ axis is the ratio between the number of predicted driver genes that intersect with genes annotated in the given hallmark gene set and the total number of genes annotated in the respective hallmark gene set. The point sizes reflect the number of driver genes playing a role in the respective hallmark gene set.

which were all TSGs (Fig 6A-C). In lung adenocarcinoma, 141 driver genes (63 OCGs and 78 TSGs) were identified by both methods (Fig 6D-F). In the case of lung adenocarcinoma, and more so than in basal-like breast cancer, the driver genes predicted by GMA were in good part also predicted by DMA.

Next, we performed enrichment analysis of the DMA predicted driver genes in basal-like breast cancer and lung adenocarcinoma to understand whether DMA and GMA can identify distinct or overlapping biological mechanisms. The significantly enriched terms (adjusted $p$-value < 0.05) among the DMA predicted driver genes in basal-like breast cancer were angiogenesis, KRAS signaling up, epithelial mesenchymal transition, and IL-2/STAT5 signaling (Fig 7A) while among the GMA predicted drivers they were IL-6/JAK/STAT3 signaling, UV response dn, KRAS signaling up and adipogenesis (Fig 3A). Thus, results from both GMA and DMA were enriched in the KRAS signaling term only.

Both GMA and DMA identified *NRP1* as a driver gene involved in KRAS signaling. *NRP1* has been shown to be highly expressed in different cancer types [64] and together with *FSTL1* is predicted to be driver for basal-like breast cancer by DMA. These two genes are involved in angiogenesis, one of the cancer hallmarks [65–67], which is also prognostic indicators of survival in breast cancer [68,69]. Additionally, for lung adenocarcinoma, key enriched terms for both DMA and GMA predicted driver genes included G2-M checkpoint, E2F targets and mTORC1 signaling (Figs 3B and 7B), suggesting that the two mechanistic indicators identify at least partially overlapping biological processes. These processes are all important in cancer progression or metastasis [70–73].

Finally, we also performed gene enrichment analysis of the driver genes identified by both DMA and GMA. An enrichment analysis of the 141 overlapping driver genes between GMA and DMA in lung adenocarcinoma revealed E2F targets, G2-M checkpoint and mTORC1 signaling to again be the most significant (Fig 7C), covering a vast majority of the overlapping genes. A similar enrichment analysis of the 13 overlapping driver genes between GMA and DMA in basal-like breast cancer revealed no significantly enriched terms.

Using methylation- and mutation-driven predictions of driver genes both have strengths and drawbacks. These two approaches are complementary as they provide a more comprehensive insight into cancer biology when used together. The quantitative nature of methylation alterations allows for an easier direct link to expression changes compared to mutations. Methylation changes capture a dynamic and reversible state, challenging their prediction as drivers, unlike mutations which are fixed changes. Additionally, methylation alterations may not always be causal and may often occur as a secondary consequence of other molecular events such as mutations. For example, methylation aberrations can often be traced back to mutations.

By integrating gene expression profiles with methylation levels, we can gain novel insights into the underlying cancer-causing mechanisms. Incorporating methylation changes provides a mechanistic explanation of the observed deregulated expression patterns and can generate new hypotheses of how epigenetics plays a role in oncogenic pathways. For instance, following prediction of methylation-driven OCGs and TSGs across cancer (sub)types, it can be speculated that certain driver genes are relevant in specific cancer (sub)types. Additionally, stage-specific methylation patterns in cancer progression can be explored. Given the reversible nature of methylation, some methylation-driven drivers might be active in specific stages of cancer development. These hypotheses can be investigated with the new GMA function.

## Availability and future directions

The data that support the findings of this study are openly available in The Cancer Genome Atlas (https://www.cancer.gov/tcga). The data used for this analysis are available at the Genomic Data Commons (https://portal.gdc.cancer.gov). GitHub and OSF repositories associated with this study are available at https://github.com/ELELAB/Moonlight2R, https://github.com/ELELAB/Moonlight2_GMA_case_studies, and https://osf.io/j4n8q/. Example data and vignette are available in S1 Data.

In the future, we envision incorporation of additional secondary -omics layers such as chromatin accessibility and copy number variation. Moreover, we would like to implement proteomics and single-cell RNA sequencing data as additional input data types. Finally, in the future, experimental studies are needed to validate the key driver genes.

## Supporting information

**S1 Fig. Integration of Moonlight and EpiMix for prediction of cancer driver genes. (A)** Heatmap showing number of differentially methylated CpGs and classifications of methylation status in the oncogenic mediators in lung adenocarcinoma. The heatmap was generated using the plotGMA function. **(B)** Venn diagram comparing oncogenic mediators predicted from Moonlight's primary layer with functional genes predicted from EpiMix in lung adenocarcinoma. The functional genes are genes containing differentially methylated CpG pairs whose DNA methylation state is associated with expression of the gene. Only those functional genes that contained the same methylation state in all of its associated CpGs were included in this comparison, and moreover, the dual methylation states were excluded. **(C)** Heatmap showing the effect of the predicted driver genes in lung adenocarcinoma on apoptosis and proliferation of cells. This heatmap was generated using the function plotMoonlightMet. These effects define the basis upon which the oncogenic mediators are predicted from the PRA step in Moonlight's primary layer. **(D)** Heatmap showing number of differentially methylated CpGs and classifications of methylation status in the oncogenic mediators in thyroid carcinoma. The heatmap was generated using the plotGMA function. **(E)** Venn diagram comparing oncogenic mediators predicted from Moonlight's primary layer with functional genes predicted from EpiMix in thyroid carcinoma. The functional genes are genes containing differentially methylated CpG pairs

whose DNA methylation state is associated with expression of the gene. Only those functional genes that contained the same methylation state in all of its associated CpGs were included in this comparison, and moreover, the dual methylation states were excluded. **(F)** Heatmap showing the effect of the predicted driver genes in thyroid carcinoma on apoptosis and proliferation of cells. This heatmap was generated using the function plotMoonlightMet. These effects define the basis upon which the oncogenic mediators are predicted from the PRA step in Moonlight's primary layer.
(PDF)

**S2 Fig. Number of driver gene-drug interactions.** Number of driver gene-drug interactions in **(A)** basal-like breast cancer, **(B)** lung adenocarcinoma, and **(C)** thyroid carcinoma found by querying DGIdb. The driver genes are stratified into OCGs and TSGs. The number of drug interactions is shown on the *x* axis and the driver genes are shown on the *y* axis.
(PDF)

**S1 Text. Methods of case study: Prediction of driver genes with differential methylation in basal-like breast cancer, lung adenocarcinoma, and thyroid carcinoma using Moonlight.**
(PDF)

**S2 Text. Results from driver-gene drug target analysis using new version of DGIdb.**
(PDF)

**S1 Data. Moonlight2R source code, with documentation and examples.**
(ZIP)

## Author contributions

**Conceptualization:** Mona Nourbakhsh, Yuanning Zheng, Matteo Tiberti, Olivier Gevaert, Elena Papaleo.

**Data curation:** Mona Nourbakhsh, Hongjin Chen, Subhayan Akhuli, Matteo Tiberti.

**Formal analysis:** Mona Nourbakhsh, Yuanning Zheng, Humaira Noor, Hongjin Chen, Subhayan Akhuli, Matteo Tiberti, Olivier Gevaert, Elena Papaleo.

**Funding acquisition:** Olivier Gevaert, Elena Papaleo.

**Investigation:** Mona Nourbakhsh, Hongjin Chen, Subhayan Akhuli, Elena Papaleo.

**Methodology:** Mona Nourbakhsh, Yuanning Zheng, Humaira Noor, Matteo Tiberti, Olivier Gevaert, Elena Papaleo.

**Project administration:** Elena Papaleo.

**Resources:** Elena Papaleo.

**Software:** Mona Nourbakhsh, Hongjin Chen, Subhayan Akhuli.

**Supervision:** Mona Nourbakhsh, Yuanning Zheng, Humaira Noor, Matteo Tiberti, Olivier Gevaert, Elena Papaleo.

**Validation:** Mona Nourbakhsh, Matteo Tiberti.

**Visualization:** Mona Nourbakhsh, Hongjin Chen, Subhayan Akhuli.

**Writing – original draft:** Mona Nourbakhsh, Hongjin Chen, Subhayan Akhuli, Elena Papaleo.

**Writing – review & editing:** Mona Nourbakhsh, Yuanning Zheng, Humaira Noor, Hongjin Chen, Subhayan Akhuli, Matteo Tiberti, Olivier Gevaert, Elena Papaleo.

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
