## [Decision Letter · Decision Letter 0]

25 Dec 2024

PCOMPBIOL-D-24-01980

Revealing cancer driver genes through integrative transcriptomic and epigenomic analyses with Moonlight

PLOS Computational Biology

Dear Dr. Papaleo,

Thank you for submitting your manuscript to PLOS Computational Biology. After careful consideration, we feel that it has merit but does not fully meet PLOS Computational Biology's publication criteria as it currently stands. Therefore, we invite you to submit a revised version of the manuscript that addresses the points raised during the review process.

Please submit your revised manuscript within 60 days Feb 24 2025 11:59PM. If you will need more time than this to complete your revisions, please reply to this message or contact the journal office at ploscompbiol@plos.org. Please include the following items when submitting your revised manuscript:

We look forward to receiving your revised manuscript.

Kind regards,

Hatice Ulku Osmanbeyoglu, Ph.D

Academic Editor

PLOS Computational Biology

Sushmita Roy

Section Editor

PLOS Computational Biology

**Journal Requirements:**

Please ensure that the funders and grant numbers match between the Financial Disclosure field and the Funding Information tab in your submission form. Note that the funders must be provided in the same order in both places as well. State what role the funders took in the study. If the funders had no role in your study, please state: "The funders had no role in study design, data collection and analysis, decision to publish, or preparation of the manuscript.".

**Reviewers' comments:**

Reviewer's Responses to Questions

**Comments to the Authors:**

Reviewer #1: Please check the attachment.

Reviewer #2: This manuscript presents an extension of the publication of moonlightR (Nourbaksh et al. 2023, 10.1093/bib/bbad274) where the utility of driver mutation analysis incorporating Cscape-somcatic was first introduced. In brief, moonlight has two layers. In the first layer, differential expression analysis yields candidate genes which are scored through several means and then partially labeled as oncogenic mediators. In a second layer, DMA is then applied, which essentially means that we look if there is a known driver mutation in the gene as I understood it. In this work, an alternative secondary layer is added which leverages the Bioconductor package EpiMix to identify DNA methylation changes that are linked to expression changes. This new functionality is termed gene methylation analysis (GMA) and results in a list of CpGs associated with expression changes and methylation state (hyp- or hypermethalyted). The manuscript is mostly well written but focuses mainly on technical details and to a lesser extent on methodological details needed to understand the method fully. Source code for the application case as well as the package itself is freely available on github. I feel that the added value of integrating methylation information in the package through EpiMix is only an incremental addition compared to the previous version of moonlight described in last year's paper. I have the following specific comments:

# Major

- The principles of DMA and PRA are insufficiently described in this manuscript and can only be understood if the reader is familiar with the previous manuscripts. Please offer a more accessible introduction to moonlight's concept for readers hearing about moonlight for the first time.

- The manuscript is vague on how the results of DMA and GMA are integrated. More information is found in S1 Text, which definitely needs to be integrated into the main manuscript to make this approach understandable . The description of the integration could nevertheless be a bit clearer. For example, what happens with the results that are not in agreement between DMA and GMA? Surely one of these layers is sufficient to determine a potential oncogene or tumor suppressor. Methylation and mutation information do not have to be always in agreement, there are ample examples where this is not the case.

- EpiMix is not a published or evaluated approach and insufficient details are provided here to understand how EpiMix is used. Also, the working principles behind EpiMix should be explained in more detail as this is the novel feature added in this manuscript. For example, are only promoter regions considered here? How are CpGs generally linked to genes?

- In the use case, the authors highlight that the identified driver genes are potential therapeutic drug targets but as far as I see this analysis is not a part of moonlight's functionality and thus somewhat unrelated. The authors may think about integrating such functionality.

- The application case should highlight more clearly if there was any added value of considering methylation data. Beyond a tradeoff of specificity / sensitivity w.r.t. COSMIC genes, how are the new results more relevant to the users. What new hypotheses could be generated here? The authors focus more on commonalities than on differences and unique insights which would be also relevant.

# Minor

- Supplemental material does not give information about package or software versions used for the analysis.

- I think the comparison of DMA and GMA results could be supported by Venn diagrams.

Reviewer #3: This manuscript proposed a tool called Moonlight2, which introduced a layer of Gene Methylation Analysis (GMA) to predict the cancer driver genes such as oncogenes and tumor suppressor genes. Moonlights integrates both DNA methylation data and transcriptomic data. This research further applied the GMA on three cancer types, including basal-like breast cancer, lung adenocarcinoma, and thyroid carcinoma. It showed that these epigenetically driven genes have the potential to forecast the prognosis of cancer patients as well as predict drug targets.

Overall this work seems to be innovative by including additional methylation data, which is important but less explored. The methods are clearly described, including the integration of EpiMix and CpG-gene pairs in methylation status. The figures are presented clearly and precisely. The analysis of genes such as PDGFRB and MET seems to be compelling to be considered as potential drug targets. Finally, the code and data are open sourced and publicly available.

The manuscripts may benefit by addressing the following concerns:

- The manuscript does not have a "discussion" section, which may mention the limitations of the research, such as the lack of experimental validation of key driver genes.

- Relevant to the point above, it will be helpful to further discuss the difference (specifically pros/cons) of using methylation-driven predictions of driver genes, in comparison to mutation-driven predictions of driver genes.

- This research mainly focused on three cancer subtypes. Expanding the study to include a pan-cancer analysis would strengthen a broader applicability of the tool.

**Have the authors made all data and (if applicable) computational code underlying the findings in their manuscript fully available?**

Reviewer #1: Yes

Reviewer #2: Yes

Reviewer #3: Yes

PLOS authors have the option to publish the peer review history of their article (what does this mean? ). If published, this will include your full peer review and any attached files.

**Do you want your identity to be public for this peer review?** For information about this choice, including consent withdrawal, please see our Privacy Policy .

Reviewer #1: No

Reviewer #2: **Yes: ** Markus List

Reviewer #3: No

**Figure resubmission:**
---

## [Decision Letter · Decision Letter 1]

26 Mar 2025

Dear Prof. Papaleo,

We are pleased to inform you that your manuscript 'Revealing cancer driver genes through integrative transcriptomic and epigenomic analyses with Moonlight' has been provisionally accepted for publication in PLOS Computational Biology.

Best regards,

Hatice Ulku Osmanbeyoglu, Ph.D

Academic Editor

PLOS Computational Biology

Sushmita Roy

Section Editor

PLOS Computational Biology

Reviewer's Responses to Questions

**Comments to the Authors:**

Reviewer #1: The authors have addressed all the comments, and the manuscript is self-contained.

Reviewer #2: I'd like to thank the authors for addressing my comments.

Reviewer #3: Addressed my concerns. Thank you.

**Have the authors made all data and (if applicable) computational code underlying the findings in their manuscript fully available?**

Reviewer #1: Yes

Reviewer #2: Yes

Reviewer #3: Yes

PLOS authors have the option to publish the peer review history of their article (what does this mean? ). If published, this will include your full peer review and any attached files.

**Do you want your identity to be public for this peer review?** For information about this choice, including consent withdrawal, please see our Privacy Policy .

Reviewer #1: No

Reviewer #2: **Yes: ** Markus List

Reviewer #3: No

---

## [Editor Report · Acceptance letter]

PCOMPBIOL-D-24-01980R1

Revealing cancer driver genes through integrative transcriptomic and epigenomic analyses with Moonlight

Dear Dr Papaleo,

I am pleased to inform you that your manuscript has been formally accepted for publication in PLOS Computational Biology. Your manuscript is now with our production department and you will be notified of the publication date in due course.

With kind regards,

Lilla Horvath
